# Long-Term Cardiovascular Findings in Williams Syndrome: A Single Medical Center Experience in Taiwan

**DOI:** 10.3390/jpm12050817

**Published:** 2022-05-18

**Authors:** Chung-Lin Lee, Shan-Miao Lin, Ming-Ren Chen, Chih-Kuang Chuang, Yu-Min Syu, Huei-Ching Chiu, Ru-Yi Tu, Yun-Ting Lo, Ya-Hui Chang, Hsiang-Yu Lin, Shuan-Pei Lin

**Affiliations:** 1Department of Pediatrics, MacKay Memorial Hospital, Taipei 10449, Taiwan; clampcage@gmail.com (C.-L.L.); miao1029@gmail.com (S.-M.L.); mingren44@gmail.com (M.-R.C.); b101098040@tmu.edu.tw (Y.-M.S.); g880a01@mmh.org.tw (H.-C.C.); wish1001026@gmail.com (Y.-H.C.); 2Institute of Clinical Medicine, National Yang-Ming Chiao-Tung University, Taipei 11221, Taiwan; 3Department of Rare Disease Center, MacKay Memorial Hospital, Taipei 10449, Taiwan; andy11tw.e347@mmh.org.tw; 4Department of Medicine, Mackay Medical College, New Taipei City 25245, Taiwan; 5Mackay Junior College of Medicine, Nursing and Management, Taipei 10449, Taiwan; 6Department of Medical Research, Division of Genetics and Metabolism, MacKay Memorial Hospital, Taipei 10449, Taiwan; mmhcck@gmail.com (C.-K.C.); likemaruko@hotmail.com (R.-Y.T.); 7College of Medicine, Fu-Jen Catholic University, Taipei 24205, Taiwan; 8Department of Medical Research, China Medical University Hospital, China Medical University, Taichung 40402, Taiwan; 9Department of Infant and Child Care, National Taipei University of Nursing and Health Sciences, Taipei 11219, Taiwan

**Keywords:** Williams syndrome, cardiovascular defect, supravalvular aortic stenosis, branch pulmonary stenosis, Taiwan

## Abstract

Williams syndrome (WS) is a rare genetic disorder caused by the microdeletion of chromosome 7q11.23. Cardiovascular defects (CVDs) are the leading causes of morbidity and mortality in patients with WS. The most common CVD in patients with WS is supravalvular aortic stenosis (SVAS), which recovers spontaneously similar to branch pulmonary stenosis (PS). Recently, conventional beliefs, such as SVAS improving rather than worsening in WS, have been challenged. This study thoroughly reviews the medical records of 30 patients with a molecular diagnosis of WS. We followed up these patients at Taipei MacKay Memorial Hospital from January 1999 to December 2021. The long-term outcomes of cardiovascular lesions as well as the change in peak pressure gradient in obstructive cardiovascular lesions over time were studied. Among these 30 patients, the most common cardiovascular lesion was SVAS (50.0%), followed by branch PS (36.7%). During the follow-up period, severe SVAS was aggravated (*p* = 0.021). The peak pressure gradient decreased from 38.4 to 25.3 mmHg (*p* = 0.001) in patients with branch PS. Among patients with WS, those with severe SVAS deteriorated over time, whereas those with branch PS improved on their own. In patients with WS who presented with branch PS, no disease-specific intervention was needed.

## 1. Introduction

Williams syndrome (WS) is a rare genetic disorder caused by the microdeletion of chromosome 7q11.23 [1]. It is characterized by typical facial features, intellectual disability, hypercalcemia, an outgoing and friendly personality, and cardiovascular defects (CVDs) [2,3,4,5], with a prevalence rate of approximately 1 per 10,000 live births [6]. Most cases are sporadic, but a few familial cases have been reported [7,8,9,10]. In WS, the length of the microdeletion in chromosome 7q11.23 is 1.4–1.8 Mb, and it contains approximately 26–28 genes, including the *ELN* gene. WS is often diagnosed using fluorescence in situ hybridization (FISH) with *ELN*-specific probes, multiplex ligation-dependent probe amplification, or chromosomal microarray (CMA) [1].

CVD is one of the most common symptoms in patients with WS. Congenital heart defects affect 80% of patients with WS [11,12]. Supravalvular aortic stenosis (SVAS) is the most common type of CVD in patients with WS, accounting for approximately 80% of all cases. The second most common type of CVD is pulmonary stenosis (PS) (valvular or peripheral), which affects approximately 45% of patients. Furthermore, in patients with WS, mitral valve prolapse, aorta coarctation, bicuspid aortic valve, and aortic hypoplasia are observed [1,13].

SVAS can be spontaneously improved in patients with WS, and some patients present normal echocardiographic findings during their lifetime. The conditions in some patients with SVAS exacerbate with age [14,15,16], whereas those in other patients improve spontaneously and even normalize [17]. According to a previous study, the long-term prognosis in patients with WS who presented SVAS was better in low initial pressure gradients (<20 mmHg) during infancy than in high-pressure gradients (>20 mmHg) [18]. Some studies have shown that 30–61% of patients with WS require surgery or catheter interventions [19,20]. As surgery for SVAS or more distal arch obstruction may reduce coronary perfusion pressure and cause myocardial ischemia, computed tomography (CT) scan, but not coronary angiography, should be performed before surgery to rule out any anomalies. If patients have ostium coronary lesions, coronary angiography can be extremely dangerous, resulting in several deaths [21].

Recently, conventional beliefs, such as SVAS improving rather than worsening in WS, have been challenged [17,22]. In our clinical practice, we observed the cardiac conditions in patients with WS who presented with SVAS as a conventional concept. However, in some cases, SVAS progressed, and the patients needed surgery. Thus, it is crucial to investigate the natural course of cardiovascular (CV) anomalies in patients with WS. In this study, we elucidated the natural course and long-term outcomes of CVD in patients with WS.

## 2. Materials and Methods

### 2.1. Patients

In this retrospective study, we reviewed and evaluated the medical records of 30 patients with WS at Taipei MacKay Memorial Hospital from January 1999 to December 2021. Patients were diagnosed according to specific morphology (small chin, long philtrum, upturned tip of nose, and sunken nasal bridge) with typical CV features on echocardiography, and the diagnoses were confirmed using FISH or array-based comparative genomic hybridization (aCGH). This study was conducted in accordance with the Declaration of Helsinki guidelines, was approved by the Mackay Memorial Hospital Institutional Review Board (21MMHIS109e, 1 October 2021), and was allowed to be published. Informed consent was obtained from all subjects involved in the study. Furthermore, written informed consent was obtained from the patients to publish this study.

### 2.2. Data Interpretation

The peak pressure gradient (PG) in obstructive CV lesions was analyzed over time. A peak PG of 10 mmHg was used to define vascular stenosis. According to the initial echocardiography report, we classified SVAS and branch PS into three categories: mild (PG ≥ 10 mm Hg < 30 mm Hg), moderate (PG ≥ 30 mm Hg < 50 mm Hg), and severe (PG ≥ 50 mm Hg). 

MedCalc version 20.027 (MedCalc Software Ltd., Ostend, Belgium) was used to perform the statistical analysis. To evaluate the peak PG change, a paired Student’s t-test was used to compare the initial and latest data. For a small, numbered group, the Wilcoxon signed-rank test was used. Multiple regressions were performed for the factors affecting the last peak PG in SVAS and branch PS. Statistical data were presented as mean values ± standard deviation. *p* < 0.05 was considered to be statistically significant.

## 3. Results

### 3.1. Cardiovascular Lesions in Williams Syndrome

In total, 30 patients who were diagnosed with WS between 1999 and 2021 were enrolled in this study. There were 19 male and 11 female patients. At the time of diagnosis, the median age was 11.3 (0–48.1) years. Using FISH analysis, 25 patients (83.3%) were diagnosed with WS and 5 patients (16.7%) were diagnosed with aCGH. The average length of follow-up was 5.6 (0.1–12.8) years. All 30 patients had CV lesions.

The most common CV lesion was supravalvular aortic stenosis (15 patients, 50.0%), followed by branch PS (11 patients, 36.7%). Furthermore, mitral regurgitation (MR, 10 patients, 33.3%), mitral valve prolapse (MVP; 9 patients, 30.0%), tricuspid regurgitation (TR; 7 patients, 23.3%), and atrial septal defect (ASD; 7 patients, 23.3%) were noted (Table 1). In patients with CV lesions (30 patients), 22 patients (73.3%) had more than one CV lesion. SVAS, branch PS and MVP are the most common CVDs in WS patients needed intervention [17]. However, no patients with SVAS, branch PS and MVP in our study underwent any intervention. Table 2 shows the severity of SVAS and branch PS on initial echocardiography. Table 3 and Table 4 revealed the baseline characteristics and comorbidities in SVAS and branch PS patients.

### 3.2. Natural Course of Supravalvular Aortic Stenosis and Pulmonary Arterial Stenosis

In our study, we examined 15 patients with SVAS and 11 with branch PS. At the start of the study, ≥2 echocardiographic records were available for all patients. The median number of years between the initial and most recent echocardiographic records was 5.6 (0.1–12.8). The peak PG of overall SVAS did not change significantly over time (Figure 1A). Only the initially severe SVAS group showed a significant increase in peak PG (initial, 55.4 ± 3.5 mmHg; latest, 65.3 ± 4.0 mmHg; *p* = 0.021) during the following time (Figure 1B–D). In all branch PS patients, a significant decrease was observed in peak velocity over time (initial, 38.4 ± 16.5 mmHg; latest, 25.3 ± 13.0 mmHg; *p* = 0.001) (Figure 2A). All three groups of branch PS (mild, moderate, and severe) showed a significant decline in peak PG (Figure 2B–D).

Figure 3A,B show that the initial peak PG was positively correlated with age at first echocardiogram in the cases of SVAS (Figure 3A) and branch PS (Figure 3B). However, no significant differences were observed (*p* = 0.75, SVAS; *p* = 0.48, branch PS).

Figure 4A,B show that the last degree of SVAS was positively correlated with age at last echocardiogram (Figure 4A) or the duration of follow-up (Figure 4B). However, no significant differences were noted (*p* = 0.88, the age at last echocardiogram; *p* = 0.61, the duration of follow-up).

Figure 5A,B show that the last degree of branch PS was positively correlated with age at last echocardiogram (Figure 5A) or the duration of follow-up (Figure 5B). Nevertheless, no significant differences were noted (*p* = 0.34, the age at last echocardiogram; *p* = 0.82, the duration of follow-up).

## 4. Discussion

The natural course of SVAS and branch PS in WS was presented in 1989 [23]. Other studies [16,18,24] reported findings that were similar to our study results. In patients with WS, the severity of SVAS progresses gradually, whereas branch PS shows rare progression and spontaneous improvement [16,18,23,24]. Nonetheless, studies from the Children’s Hospital of Philadelphia [17] and Taiwan [22] refuted the natural course of SVAS. SVAS in patients with WS is more likely to improve than worsen, even though the result was not statistically significant.

Similar to a study conducted in Korea [25], this was the first study in Taiwan to demonstrate that SVAS in patients with WS would gradually progress only in the initially severe SVAS group and not in the mild and moderate SVAS groups. There was no significant progression of overall SVAS in patients with WS. It was observed that all groups of patients with branch PS (mild to severe) improved spontaneously during the follow-up periods. The natural course of CV lesions in WS was usually followed using catheterization data in previous studies [16,18,23,24]; therefore, it was novel to follow the natural course of SVAS and branch PS using echocardiographic data. Catheterization for the diagnosis of CV lesions in patients with WS was replaced with echocardiography because of its invasiveness.

Our study revealed that it was unnecessary to treat branch PS using balloon or surgical angioplasty in patients with WS. No patients with branch PS had undergone balloon or surgical angioplasty in the present study. According to the conventional therapeutic concept, patients with severe branch PS in WS require balloon pulmonary angioplasty at first. Moreover, 50% of patients with branch PS had a disease-specific intervention, with catheter-based intervention being the most common [17]. Another study found that only 4.7% of patients with branch PS required disease-specific treatment [25]. Although the group of branch PS in WS was initially severe, it improved over time. Most patients with branch PS in WS did not require intervention according to the intervention-free probability.

The age at initial examination should be considered when conducting statistical analyses. In our study, many patients were followed up at our outpatient department one year after cardiac manifestation. However, the evolution of lesions is more dynamic in the first year of life. In our study, the degree of SVAS and branch PS were positively correlated with the age at the initial examination. Nevertheless, there were no significant differences.

MVP was another lesion that needed intervention. Our study patients with MVP did not undergo any intervention; however, a previous study revealed that mitral valve surgery for MVP was the second most common surgical intervention, which showed favorable results after surgery [25]. In Argentina, a study reported four cases of surgical intervention for MVP [26]. Despite this, most previous studies [25,27,28] found no need for mitral surgery in WS due to the lower severity of MVP in WS [27]. According to Cha et al. [25], 33% of patients with MR in WS were moderate to severe, accompanied by left ventricular and left atrial enlargement. Thus, performing mitral valve surgery was an unavoidable option.

Based on previous studies, ascending aortoplasty is the most common surgical intervention for SVAS [17,19,25,26]. Irrespective of the type of surgical intervention, ascending aortoplasty showed good results in WS. None of the patients needed reoperation. Despite undergoing several surgical techniques, patients showed minimal SVAS after the operation. According to the results of the intervention-free probability in SVAS, the risk of intervention in the long-term follow-up was related to the severity of SVAS at the initial echocardiography [17,25].

Hypertension is the most prevalent nonstructural cardiovascular issue in patients with WS. The etiology of hypertension in patients with WS remains unknown. Renal artery stenosis is considered a possible cause of hypertension in patients with WS, with a prevalence rate of 7–58% [28,29]. In our study, 20% patients were diagnosed with hypertension at last evaluation; however, only 16.7% of these 20% patients had renal artery stenosis.

Our study has some limitations. First, the research data originated from a single medical center. Some medical centers used peak velocity without PG as an echocardiography report for the evaluation of SVAS and branch PS. This leads to a selection bias. Second, this was a retrospective study. Some patients had only one echocardiogram record during their follow-up, and each patient had an irregular interval between the echocardiogram records. This factor could also influence the selection bias. The follow-up duration in our patients was between 0.1 and 12 years, which would make the statistical analysis heterogeneous. To handle this problem, the interval between echocardiogram records should be controlled and it should be nearly the same between each echocardiogram.

Furthermore, it is extremely difficult to trace the natural progression of cardiovascular diseases in patients with WS from the time of birth according to a previous study [17]. Most patients with WS had undergone cardiology evaluations before they were diagnosed with WS in real-world practice, as per previous studies [17,28].

## 5. Conclusions

Only the initially severe SVAS group exacerbated with time among patients with WS, whereas all groups of branch PS improved spontaneously, including the severe group. If patients had branch PS, most of them did not require disease-specific interventions. Thus, an experienced multidisciplinary team is needed to provide the necessary CV care and follow-up.

## Figures and Tables

**Figure 1 jpm-12-00817-f001:**
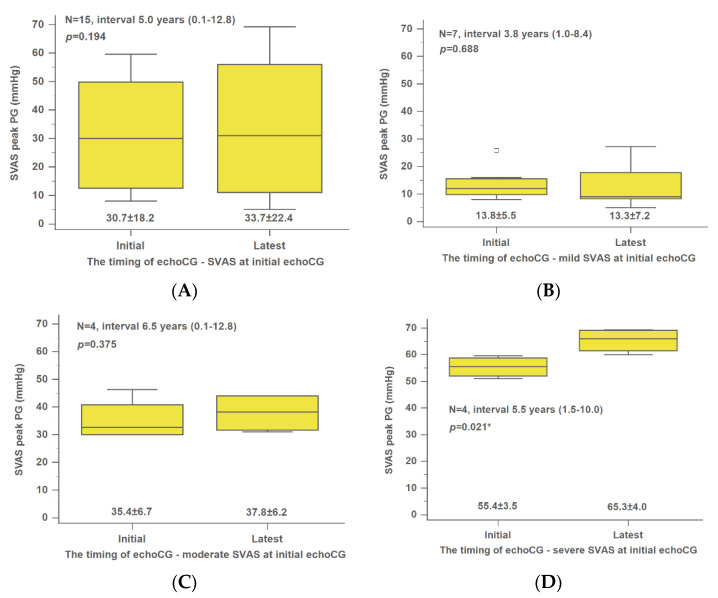
Changes in the mean peak velocity of overall (**A**), mild (**B**), moderate (**C**), and severe (**D**) SVAS at initial echocardiography. The interval was presented as the median value. echoCG, echocardiography; PG, pressure gradient; SVAS, supravalvular aortic stenosis. * *p*-value was calculated using the Wilcoxon signed-rank test.

**Figure 2 jpm-12-00817-f002:**
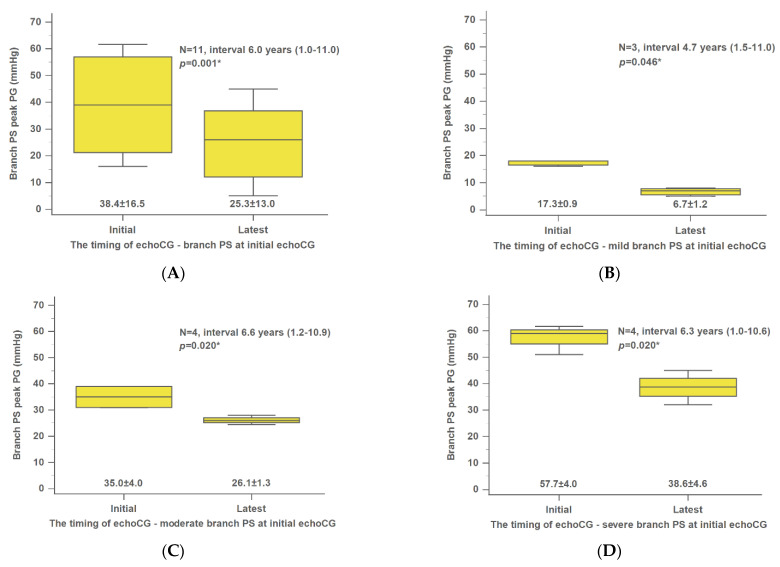
Changes in the mean peak velocity of overall (**A**), mild (**B**), moderate (**C**), and severe (**D**) branch PS at the initial echocardiography. The interval was presented as the median value. echoCG, echocardiography; PG, pressure gradient; PS, pulmonary stenosis. * *p*-value was calculated using the Wilcoxon signed-rank test.

**Figure 3 jpm-12-00817-f003:**
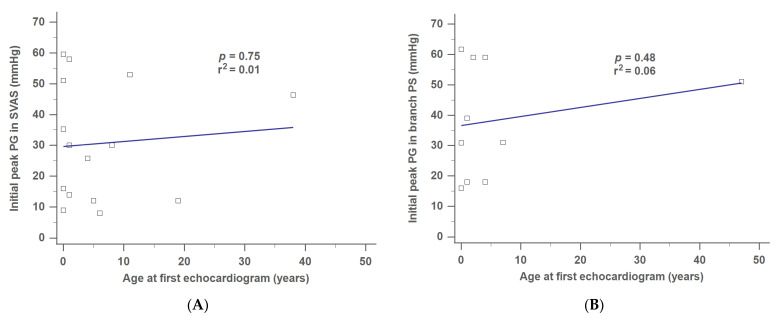
Correlation between the initial peak PG and age at first echocardiogram in (**A**) SVAS, and (**B**) branch PS. PG, pressure gradient; SVAS, supravalvular aortic stenosis; PS, pulmonary stenosis.

**Figure 4 jpm-12-00817-f004:**
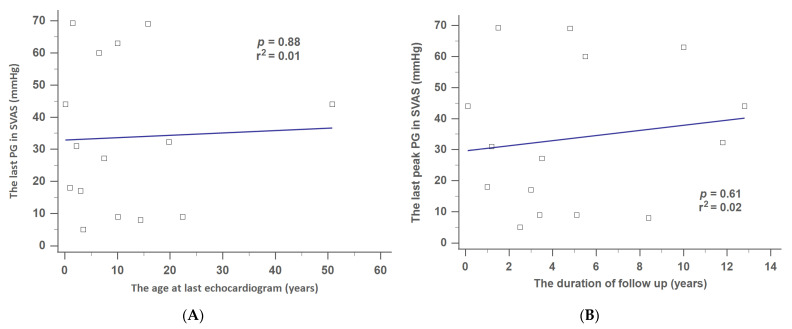
Correlation between the last peak PG and (**A**) age at last echocardiogram, or (**B**) duration of follow-up in SVAS. PG, pressure gradient; SVAS, supravalvular aortic stenosis.

**Figure 5 jpm-12-00817-f005:**
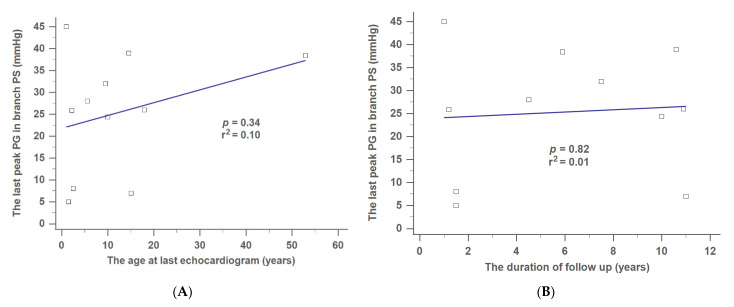
Correlation between the last peak PG and (**A**) age at last echocardiogram, or (**B**) duration of follow-up in branch PS. PG, pressure gradient; PS, pulmonary stenosis.

**Table 1 jpm-12-00817-t001:** Number and percentage of cardiovascular lesions in Williams syndrome.

Cardiovascular Lesions	Number of Patients (%) (*n* = 30)
Supravalvular AS	15 (50.0%)
Branch PS	11 (36.7%)
MR	10 (33.3%)
Mitral valve prolapse	9 (30.0%)
TR	7 (23.3%)
ASD	7 (23.3%)
PLSVC	3 (10.0%)
Right pulmonary artery stenosis	3 (10.0%)
Tricuspid valve prolapse	2 (6.7%)
Increased AsAo	2 (6.7%)
Supravalvular pulmonary stenosis	1 (3.3%)
VSD	1 (3.3%)
Small AsAo	1 (3.3%)
Small DsAo	1 (3.3%)
MS	1 (3.3%)
Dilated Ao root	1 (3.3%)

Abbreviations: Ao, aortic; AS, aortic stenosis; AsAo, ascending aorta; ASD, atrial septal defect; DsAo, descending aorta; MR, mitral regurgitation; MS, mitral stenosis; PLSVC, persistent left superior vena cava; PS, pulmonary stenosis; TR, tricuspid regurgitation; VSD, ventricular septal defect.

**Table 2 jpm-12-00817-t002:** Severity groups of SVAS and branch PS.

Cardiovascular Lesions	Mild	Moderate	Severe
SVAS	7/15 (46.6%)	4/15 (26.7%)	4/15 (26.7%)
Branch PS	3/11 (27.2%)	4/11 (36.4%)	4/11 (36.4%)

Abbreviations: SVAS, supravalvular aortic stenosis; PS, pulmonary stenosis.

**Table 3 jpm-12-00817-t003:** The baseline characteristics and comorbidities in SVAS.

Patient Number	Gender	Age at First Echocardiography (Year-Old)	Age at Last Echocardiography (Year-Old)	Duration of Follow Up (Years)	Initial Peak PG (mmHg)	Latest Peak PG (mmHg)	Other CVDs
1	male	8	19.8	11.8	30	32.3	MVP
2	male	11	15.8	4.8	53	69	MR
3	female	0	10.0	10.0	51	63	Branch PS, AS, Right pulmonary artery stenosis
4	male	19	22.4	3.4	12	9	None
5	male	4	7.5	3.5	25.8	27.2	None
6	female	38	50.8	12.8	46.3	44	MVP
7	male	0	1.5	1.5	59.6	69.3	Branch PS, Right pulmonary artery stenosis
8	male	5	10.1	5.1	12	9	None
9	male	1	6.5	5.5	58	60	None
10	male	1	3.5	2.5	14	5	Increased AsAo
11	female	1	2.2	1.2	30	31	Branch PS, MR, TR
12	male	0	3.0	3.0	16	17	None
13	female	6	14.4	8.4	8	8	MVP, MR
14	male	0	0.1	0.1	35.3	44	ASD, PLSVC
15	male	0	1.0	1.0	9	18	Branch PS, ASD

Abbreviations: AsAo, ascending aorta; ASD, atrial septal defect; CVDs, cardiovascular defects; MR, mitral regurgitation; MVP, mitral valve prolapse; PLSVC, persistent left superior vena cava; PS, pulmonary stenosis; SVAS, supravalvular aortic stenosis; TR, tricuspid regurgitation.

**Table 4 jpm-12-00817-t004:** The baseline characteristics and comorbidities in branch PS.

Patient Number	Gender	Age at First Echocardiography (Year)	Age at Last Echocardiography (Year)	Duration of Follow Up (Years)	Initial Peak PG (mmHg)	Latest Peak PG (mmHg)	Other CVDs
1	female	0	10.0	10.0	30.9	24.4	SVAS, Right pulmonary artery stenosis
2	male	4	14.6	10.6	59	39	MVP
3	male	7	17.9	10.9	31	26	TR
4	male	0	1.5	1.5	16	5	SVAS, Right pulmonary artery stenosis
5	female	2	9.5	7.5	59	32	MVP, MR, TR, Small AsAo, Small DsDo
6	female	47	52.9	5.9	51	38.4	MR, TR
7	female	1	5.5	4.5	39	28	MVP, MR, PLSVC, TVP, TR, MS
8	female	1	2.5	1.5	18	8	Right pulmonary artery stenosis
9	male	4	15.0	11.0	18	7	MVP, MR, TR
10	female	1	2.2	1.2	39	25.9	SVAS, MR, TR
11	male	0	1.0	1.0	61.7	45	SVAS, ASD

Abbreviations: AsAo, ascending aorta; ASD, atrial septal defect; CVDs, cardiovascular defects; DsAo, descending aorta; MR, mitral regurgitation; MS, mitral stenosis; MVP, mitral valve prolapse; PLSVC, persistent left superior vena cava; PS, pulmonary stenosis; SVAS, supravalvular aortic stenosis; TR, tricuspid regurgitation; TVP, tricuspid valve proplase.

## Data Availability

All data are present within the article.

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
