# Peer review of "Long-Term Cardiovascular Findings in Williams Syndrome: A Single Medical Center Experience in Taiwan"

_jpm, 2022, doi:10.3390/jpm12050817_

Round 1

Reviewer 1 Report

No further comments

Authors did their best to improve the manuscripts. It remains unclear what the graphical analysis add in this present form

Author Response

1. No further comments. Authors did their best to improve the manuscripts. It remains unclear what the graphical analysis add in this present form.

=> Thank you for your suggestion. Figure 3A and 3B showed that the initial peak PG of SVAS and PS were positively correlated with the age at the first echocardiogram; however, there were no significant differences. Figure 4A and 4B showed that the age at last echocardiogram and the duration of follow up were positively correlated with the last degree of SVAS; nevertheless, there were no significant differences. Figure 5A and 5B showed that the age at last echocardiogram and the duration of follow up were positively correlated with the last degree of PS;  however, no significant differences were observed.

Reviewer 2 Report

The Authors investigated the long-term outcomes of aortic stenosis and branch pulmonary stenosis  in patients with molecularly assessed Williams Syndrome threated and in follow up at their tertiary center.

The topic is interesting and the paper is well-written. The Authors retrospectively surveyed and analyzed the cardiovascular lesions of WS-Patients and report the changes at mean peak velocity at 0.1-12 years follow-up. The study cohort is small and the methods are not elaborate, but the results for this rare syndrom are useful in the clinical practice.

Title: appropriate.

Abstract: Informative. In line with the main text.

Key Points: exhaustive.

Keywords: appropriate.

Informed Consent: appropriate

Ethical Committee: appropriate.

References: Satisfactory.

Figures and tables: Satisfactory.

Specific Comments

Introduction

The introduction is very comprehensive, and the topic as well as the previous literature about the theme is adequately illustrated.

The abbreviation CT and CV have not been introduced, Please introduce and write in the extensive form, befor using the abbreviation.

Methods

The echographic peak pressure gradient measurement  describe in the methods would warrant a separate paragraph.

Results

In the discussion (Page 8, line 187) patients with MPV not needing intervention are cited. Please include this information in your results.  I suggest to specify other comorbidities and baseline characteristics of your patients, if available in the data.

Discussion

appropriate.

Figure 4

Layout correction needed in line 147   

Author Response

1.Introduction

The introduction is very comprehensive, and the topic as well as the previous literature about the theme is adequately illustrated.

The abbreviation CT and CV have not been introduced, Please introduce and write in the extensive form, before using the abbreviation.

=> Thank you for your suggestion. We have  introduced the abbreviation CT and CV in the latest manuscript. (Page 4, Line 100; Page 5, Line 108)

2. Methods

The echographic peak pressure gradient measurement describe in the methods would warrant a separate paragraph.

=> Thank you for your suggestion. We have  separated the echographic peak pressure gradient measurement description as a single paragraph in the latest manuscript. (Page 5, Line 125)

3. Results

In the discussion (Page 8, line 187) patients with MPV not needing intervention are cited. Please include this information in your results. I suggest to specify other comorbidities and baseline characteristics of your patients, if available in the data.

=> Thank you for your suggestion. The discussion of MVP was described in our results in the latest manuscript (Page 6, Line 150. “SVAS, branch PS and MVP are the most common CVDs in WS patients needed intervention [17]. However, no patients with SVAS, branch PS and MVP in our study underwent any intervention.”). The comorbidities and baseline characteristics of our patients were revealed in Table 3 and Table 4.

4. Figure 4

Layout correction needed in line 147  

=> Thank you for your suggestion. The layout of figure 4 was corrected in the latest manuscript.

This manuscript is a resubmission of an earlier submission. The following is a list of the peer review reports and author responses from that submission.

Round 1

Reviewer 1 Report

Line 144 and 145 say that the severe SVAS group had an increase in peak pressure gradient, as shown in Fig 1D, but Fig 1D doesn't reflect the values mentioned in the text. Also check the figure label for 1D.

Spell check at line 196.

Author Response

1. Line 144 and 145 say that the severe SVAS group had an increase in peak pressure gradient, as shown in Fig 1D, but Fig 1D doesn't reflect the values mentioned in the text. Also check the figure label for 1D.

=>Thank you for your suggestion. We have the wrong figure in Fig 1D, and we would modify the figure to the correct figure.

2. Spell check at line 196.

=> Thank you for your suggestion. We have modified the spell at line 196.

Reviewer 2 Report

In this retrospective single centre cohort study of 30 patients with Williams Beuren syndrome, the authors reviewed the evolution of supravalvular aortic and pulmonary stenosis. 

They observed that patients with severe aortic stenosis deteriorated over time, while all branch pulmonary stenosis improved. 

As acknowledged,  the study had some limitations inherent to the retrospective single centre design, low number of patients, short and heterogeneous follow-up . 

The statistical analysis compared gradient evolution as a mean data from 0.1 to 12 years of follow up  making it very confusing and  heterogeneous. 

The statistical analysis seems not to take in consideration the age at the initial exam.  Indeed evolution of lesions may be more dynamic in the first years of life.

I would suggest reformatting  the graph 

Given the low number of patients   a point for gradient should be depicted per patient for each exam and according to the age of the patients.  There fore a curve of the evolution will be depicted for each patient depending on the age , gradient and duration of follow up   then a regression analysis could give the overall trend of the cohort  over the age 

Unusual writing  is observed in some instances.  The abstract lacks conclusion.   "Recent research, however, has refuted this theory "   please reformulate

introduction :  too long  should be synthetized and more focused on why this study was performed. 

methods :   Ethical approval ?   Consent of patients or legal representatives ?

Why a single centre cohort ?  Sould it be possible to have a more collaborative work increasing study efficiency

It is stated that coronary angiography should be performed.  I would be very cautious with this statement   given sometimes ostia coronary lesions , this exam can be very dangerous and death have been reported (at least in congress) .   CT scan may be the first line exam 

Author Response

1. The statistical analysis compared gradient evolution as a mean data from 0.1 to 12 years of follow up making it very confusing and heterogeneous.  

=> Thank you for your suggestion. We would remove this mention to make our statistical analysis clear.

2. The statistical analysis seems not to take in consideration the age at the initial exam. Indeed, evolution of lesions may be more dynamic in the first years of life.

=> Thank you for your suggestion. This is the limitation of our study. Most patients were followed up at our OPD after 1-year-old with cardiac manifestation. We would mention this limitation in our article. “The statistical analysis should be taken in consideration the age at the initial examination. In our study, a large proportion of patients were followed up at our OPD after 1-year-old with cardiac manifestation. However, evolution of lesions may be more dynamic in the first years of life.” at line 250.

3. Given the low number of patients a point for gradient should be depicted per patient for each exam and according to the age of the patients. Therefore, a curve of the evolution will be depicted for each patient depending on the age, gradient and duration of follow up then a regression analysis could give the overall trend of the cohort over the age.

=> Thank you for your suggestion. We would give the regression analysis in Table 3 and Table 4 for the overall trend of the cohort over the age. “Table 3 showed the factors affecting the last peak PG in SVAS. There was positive correlation between the degree of SVAS and age but negative correlation between the degree of SVAS and duration of follow up. However, they had no significant differences (p=0.606 in age and p=0.939 in duration of follow up). Table 4 showed the factors affecting the last peak PG in branch PS. There was positive correlation between the degree of branch PS and age. The positive correlation between the degree of branch PS and duration of follow up was also noted. Nevertheless, no significant differences were noted (p=0.680 in age and p=0.955 in duration of follow up).” at line 179.

4. Unusual writing is observed in some instances. The abstract lacks conclusion. "Recent research, however, has refuted this theory " please reformulate.

=> Thank you for your suggestion. We would add the conclusion in abstract. “In WS, the severe SVAS group deteriorated over time, while all branch PS groups improved on their own. In patients with WS with branch PS, there was no need for disease-specific intervention” at line 66. And we would modify this sentence “Recent research, however, has refuted this theory” to “In recent studies, conventional beliefs such as SVAS improving rather than worsening in WS were challenged.” at line 57.

5. Introduction: Too long. It should be synthetized and more focused on why this study was performed.

=> Thank you for your suggestion. We have canceled some paragraphs and synthetized the introduction at line 87.

6. Methods: Ethical approval? Consent of patients or legal representatives?

=> Thank you for your suggestion. Ethical approval and consent of patients were presented at line 141. “This study was carried out in accordance with the Declaration of Helsinki guidelines, was approved by the Mackay Memorial Hospital Institutional Review Board (21MMHIS109e, 2021/10/01), and was allowed to be published. Informed consent was obtained from all subjects involved in the study. Written informed consent has been obtained from the patients to publish this paper.”

7. Why a single center cohort? Should it be possible to have a more collaborative work increasing study efficiency

=> Thank you for your suggestion. This is the limitation of our study. Some medical center used peak velocity without PG as echocardiography report for evaluation of SVAS and branch PS. We would mention this issue in limitation part. “Some medical center used peak velocity without PG as echocardiography report for evaluation of SVAS and branch PS.” at line 250.

8. It is stated that coronary angiography should be performed. I would be very cautious with this statement that given sometimes ostium coronary lesions, this exam can be very dangerous, and death have been reported (at least in congress). CT scan may be the first line exam

=> Thank you for your suggestion. We would modify this statement. “Because surgery for SVAS or more distal arch obstruction could reduce coronary perfusion pressure and cause myocardial ischemia, CT scan but not coronary angiography should be performed before surgery to rule out any anomalies. If patients have ostium coronary lesions, coronary angiography can be very dangerous, and death have been reported” at line 119.

Round 2

Reviewer 1 Report

Accepted

Author Response

Thank you for your review.

Reviewer 2 Report

The authors have made some revisions of the manuscript but it still need a strong effort of reformating. 

Introduction remains too long  and more written as a review than as a real introduction.    The real context of the studies with discrepancies and challenge in recent studies  should be a little more depicted in the introduction to introduce the controversies  while many clinical details  could be removed. 

Despite i am not a native english speaker, i believe the paper should have a substantial revision with regards to syntax. 

1. The statistical analysis compared gradient evolution as a mean data from 0.1 to 12 years of follow up making it very confusing and heterogeneous.

=> Thank you for your suggestion. We would remove this mention to make our statistical analysis clear.

=>  Reviewer   i do not suggest to remove this important information that is a major limit of the study. It should even be made clearer in the discussion  and  a discussion with regards to how to handle this limit should be made. The present graphs are not acceptable

2. The statistical analysis seems not to take in consideration the age at the initial exam. Indeed, evolution of lesions may be more dynamic in the first years of life.

No result are provided to answer this comment. 

3. Given the low number of patients a point for gradient should be depicted per patient for each exam and according to the age of the patients. Therefore, a curve of the evolution will be depicted for each patient depending on the age, gradient and duration of follow up then a regression analysis could give the overall trend of the cohort over the age.

=> graphs were not produced in the revision  and are expected

Author Response

1.Introduction remains too long and more written as a review than as a real introduction. The real context of the studies with discrepancies and challenge in recent studies should be a little more depicted in the introduction to introduce the controversies while many clinical details could be removed.

=>Thank you for your suggestion. We have removed clinical details and focused on  discrepancies and challenge in recent studies.

2. The statistical analysis compared gradient evolution as a mean data from 0.1 to 12 years of follow up making it very confusing and heterogeneous.

=> I do not suggest to remove this important information that is a major limit of the study. It should even be made clearer in the discussion and a discussion with regards to how to handle this limit should be made. The present graphs are not acceptable

=>Thank you for your suggestion. We have made it clearer in the discussion and a discussion with regards to how to handle this limit should be made. “ The following years of our patients were between 0.1 to 12 years. It would make the statistical analysis heterogeneous. To handle this problem, the interval of echocardiogram should be controlled as nearly the same when we make echocardiogram.” at page 10, line 244. 

3. The statistical analysis seems not to take in consideration the age at the initial exam. Indeed, evolution of lesions may be more dynamic in the first years of life.

=> No result are provided to answer this comment.

=>Thank you for your suggestion. We have analyzed the correlation of the initial peak PG and the age at first echocardiogram in SVAS and branch PS “ Figure 3A and 3B revealed the correlation of the initial peak PG and the age at first echocardiogram in SVAS (Figure 3A), and branch PS (Figure 3B). There were positive correlation between the initial peak PG and the age at first echocardiogram in SVAS and branch PS. However, no significant differences were noted (p=0.5727 in SVAS and p=0.4843 in branch PS)” at page 7, line 159. “ The statistical analysis should be taken in consideration the age at the initial examination. In our study, a large proportion of patients were followed up at our OPD after 1-year-old with cardiac manifestation. However, evolution of lesions may be more dynamic in the first years of life. In our study, the degree of SVAS and branch PS were positive correlation with the age at the initial examination. Nevertheless, there was significant differences.” at page 9, line 205.

4. Given the low number of patients a point for gradient should be depicted per patient for each exam and according to the age of the patients. Therefore, a curve of the evolution will be depicted for each patient depending on the age, gradient and duration of follow up then a regression analysis could give the overall trend of the cohort over the age.

=> Graphs were not produced in the revision and are expected.

=>

Thank you for your suggestion. We have produced the graphs in figure 4A,4B,5A,5B.

“ Table 3 showed the factors affecting the last peak PG in SVAS. There was positive correlation between the last degree of SVAS and the age at last echocardiogram. The positive correlation was also noted between the last degree of SVAS and the duration of follow up. However, they had no significant differences (p=0.8839 in the age at last echocardiogram and p=0.6127 in the duration of follow up). Figure 4A and 4B reflected the correlation of the last degree of SVAS and the age at last echocardiogram (Figure 4A) or the duration of follow up (Figure 4B).” at page 7, line 164.

“ Table 4 showed the factors affecting the last peak PG in branch PS. There was positive correlation between the last degree of branch PS and the age at last echocardiogram. The positive correlation was also noted between the last degree of branch PS and the duration of follow up. Nevertheless, no significant differences were noted (p=0.3308 in the age at last echocardiogram and p=0.8237 in the duration of follow up). Figure 5A and 5B reflected the correlation of the last degree of branch PS and the age at last echocardiogram (Figure 5A) or the duration of follow up (Figure 5B).” at page 7, line 171.
